# Biocatalytic Silylation: The Condensation of Phenols and Alcohols with Triethylsilanol



Emily I. Sparkes [1,2], Chisom S. Egedeuzu [1,2], Billie Lias [1,2], Rehana Sung [1], Stephanie A. Caslin [1,2], S. Yasin Tabatabaei Dakhili [1,2], Peter G. Taylor [3], Peter Quayle [2] and Lu Shin Wong [1,2,*]

[1] Manchester Institute of Biotechnology, University of Manchester, 131 Princess Street, Manchester M1 7DN, UK; emilysparkes11@gmail.com (E.I.S.); chisom.egedeuzu@manchester.ac.uk (C.S.E.); billie.lias@student.manchester.ac.uk (B.L.); rehana.sung@manchester.ac.uk (R.S.); stephanie.caslin@sky.com (S.A.C.); s.yasin.tabatabaei.d@gmail.com (S.Y.T.D.)
[2] Department of Chemistry, University of Manchester, Oxford Road, Manchester M13 9PL, UK; Peter.Quayle@manchester.ac.uk
[3] Faculty of Science, Technology, Engineering and Mathematics, Open University, Walton Hall, Milton Keynes MK7 6AA, UK; peter.taylor@open.ac.uk
* Correspondence: l.s.wong@manchester.ac.uk

**Abstract:** Silicatein-$\alpha$ (Sil$\alpha$), a hydrolytic enzyme derived from siliceous marine sponges, is one of the few enzymes in nature capable of catalysing the metathesis of silicon–oxygen bonds. It is therefore of interest as a possible biocatalyst for the synthesis of organosiloxanes. To further investigate the substrate scope of this enzyme, a series of condensation reactions with a variety of phenols and aliphatic alcohols were carried out. In general, it was observed that Sil$\alpha$ demonstrated a preference for phenols, though the conversions were relatively modest in most cases. In the two pairs of chiral alcohols that were investigated, it was found that the enzyme displayed a preference for the silylation of the *S*-enantiomers. Additionally, the enzyme's tolerance to a range of solvents was tested. Sil$\alpha$ had the highest level of substrate conversion in the nonpolar solvents *n*-octane and toluene, although the inclusion of up to 20% of 1,4-dioxane was tolerated. These results suggest that Sil$\alpha$ is a potential candidate for directed evolution toward future application as a robust and selective biocatalyst for organosiloxane chemistry.

**Keywords:** silicatein; condensation; silyl ether; organosiloxanes; biocatalysis





## 1. Introduction

During the multi-step chemical synthesis of complex molecules, silyl ethers are often employed for the protection of hydroxyl groups [1–7], where their utility arises from orthogonality to other commonly used acid- and base-labile protecting groups. Typically, the introduction of these silyl groups involves the use of an electrophilic silylating reagent such as a silyl chloride (i.e., chlorosilane) or triflate, the latter of which is itself produced from the corresponding silyl chloride [8]. The trimethylsilylation of alcohols has also been demonstrated with *N,O*-bis-silyl trifluoroacetamide [9], *N,N'*-bis-silyl urea [10] and hexamethydisilazane [11]. It has also been known for many years that the silylation of alcohols can be affected by hydrosilanes (silyl hydrides), with the dehydrogenative condensation catalysed by transition metals or strong Lewis acids [12–14]. Alternatively, silylation can also be affected by activation of the silanol to attack by an alcohol (or phenol) under Mitsunobu conditions [15].

However, in all cases, the necessary reagents are energy intensive to produce, and their use results in the generation of stoichiometric amounts of by-products that are hazardous or environmentally undesirable (e.g., triflic acid, hydrogen chloride, hydrogen). In this regard, the capability of silylate hydroxy groups through the condensation of the corresponding silanol and alcohol would circumvent the need for harsh reagents and only release water

as the by-product. However, only one example of this dehydrative condensation has been reported, catalysed by rare-earth Lewis acids such as Yb(OTf)$_3$ and Sc(OTf)$_3$ [16]. It would therefore be preferable to identify a reaction pathway that better conforms to the principles of green chemistry—in particular, the harnessing of biological catalysts that can be sustainably sourced and avoid any requirement for rare metals [17].

Silicatein-$\alpha$ (Sil$\alpha$), an enzyme responsible for the condensation of inorganic Si–O bonds in marine demosponges [18,19], has also previously been shown to catalyse the condensation of a variety of organosilanols with aliphatic alcohols to produce the corresponding silyl ether [20]. This enzyme is monomeric and does not require any co-factors for activity, so may offer a relatively benign alternative biocatalytic approach for silylation. However, the general substrate scope of this enzyme is currently not well established and requires further investigation before synthetically useful methods can be subsequently developed.

In this study, the substrate scope from the perspective of the alcohol component was investigated by surveying the enzyme's ability to form triethylsilyl ethers with a range of phenols and aliphatic alcohols. In addition, Sil$\alpha$'s tolerance for a variety of polar and nonpolar solvents was tested to examine the optimum reaction conditions for Sil$\alpha$ in organic media. In the original report of Sil$\alpha$-catalysed condensations, a preference for phenolic hydroxy groups was found [20]. However, more recent studies found that the hexahistidine affinity tag used for isolation of the enzyme was also contributing to nonspecific catalysis [21]. Thus, in order to more accurately determine the substrate scope and catalysis mediated by the enzyme active site itself, in this study we used an enzyme construct consisting of the ribosomal chaperone protein trigger factor fused to the *N*-terminus and a Strep II tag fused to the *C*-terminus (henceforth referred to as TF-Sil$\alpha$-Strep).

## 2. Results and Discussion

### 2.1. Triethylsilylation of Phenols

The triethylsilylation of various substituted phenols was first examined to investigate the effect of the substituents on the enzyme-catalysed reaction. Chloro, methyl and methoxy substituents were chosen to represent electron-donating and -withdrawing groups, and all three positions (*ortho*, *meta* and *para*) were investigated. The condensation reactions were carried out using the procedure previously reported in [20], whereby the enzyme was used in the form of a lyophilised solid (in a matrix of potassium salts and 18-crown-6) in organic solvent at 75 °C (Scheme 1). The product conversions were measured after 72 h by GC-MS.

**Scheme 1.** General reaction scheme for the condensation of the alcohol with triethylsilanol (TES-OH) to form the corresponding silyl ether. R: aliphatic or substituted phenyl groups.

For all phenols tested, some degree of product formation was observed, even in control reactions where the enzyme was omitted. However, in all cases, the inclusion of Sil$\alpha$ resulted in enhanced conversions, though some were deemed not to be statistically significant (Figure 1). Unsubstituted phenol was well accepted by the enzyme, giving a gross conversion of 88% after 72 h and a net conversion of 61% after subtraction of the percentage conversion from the nonenzymatic reaction (Table S1 in the Supplementary Information). High net conversions were also observed with *p*-methoxyphenol at 62% (75% gross). This result equated to a nearly 6-fold higher conversion for the enzyme-catalysed reaction vs. the nonenzymatic reaction (i.e., 75% vs. 13%). *o*-Methoxyphenol was the least reactive substrate under these conditions, giving gross and net conversions of 15% and 11%, respectively. Even so, this result still represented a 3.7-fold increase in conversion, which is attributable to the enzyme because the control reaction gave very little conversion.

In all cases, the remaining material was the unconverted alcohol, silanol and the disiloxane self-condensation product.

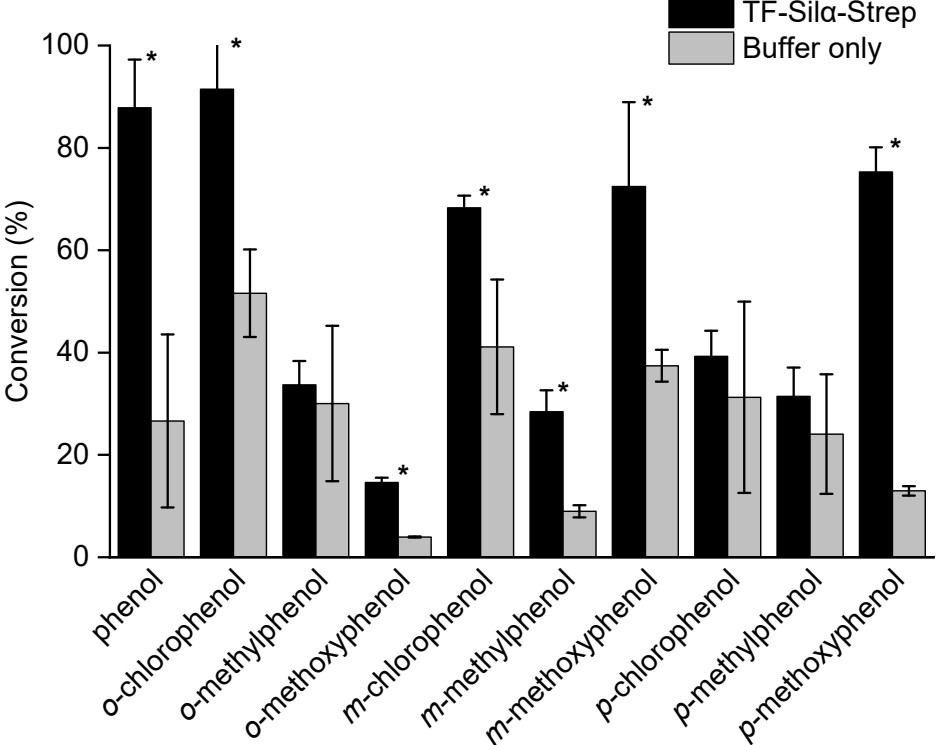

**Figure 1.** Graph of percentage conversions of phenols to the corresponding silyl ethers after 72 h. The error bars indicate standard deviations. A one-tailed Student's *t*-test assuming unequal variance was performed, comparing each enzyme to its control. Comparisons resulting in a $p < 0.05$ were deemed to be significant and are marked with *.

In comparing the *ortho*-substituted substrates, 3.7- and 1.8-fold enhancements in the conversion were observed for the methoxy- and chlorophenols, respectively. There was no significant difference in conversion for *o*-methylphenol compared to its control. Since the methoxy-bearing substrate is the most sterically demanding (i.e., volume occupied as quantified by their ligand repulsive energies [22]), its preference could be due to favourable electronic interactions with the enzyme such as hydrogen bonding or dipole–dipole interactions. Similar interactions could also be responsible for the enhancement observed with *o*-chlorophenol.

The phenols with the *meta*-substituents all exhibited enzymatic enhancements to their conversion that were statistically significant, but no trends could be identified. The greatest degree of enzymatic enhancement was found with *m*-methylphenol with a 3.2-fold improvement over the control, though the net percentage conversion was low at just 19%. The methoxy- and chloro-substituted phenols gave higher absolute conversions, but lower levels of enzymatic enhancement at 1.9- and 1.7-fold respectively, since the control reactions gave relatively high conversions even in the absence of the enzyme.

Of the three *para*-substituted phenols, only *p*-methoxyphenol showed a statistically significant conversion enhancement. Indeed, the enzyme gave the highest conversion improvement amongst all the tested phenols (Figure 1). Since this substrate presents the bulkiest group of the three *para*-substituted phenols, it is unlikely that this preference is due to steric factors and suggests that the molecule is forming specific interactions that are favourable to either binding or catalysis.

From the perspective of the type of substituent (Figure S1 and Table S1 in Supplementary Information), the chlorophenols generally gave higher conversions for both enzyme-

catalysed and uncatalysed reactions, and thus the lowest fold improvements (<1.8 in all cases). Conversely, the methoxyphenols gave low conversions in the control reactions, but were the most improved by the addition of the enzyme with fold increases ranging from 1.9 to 5.8. The cresols (methylphenols) showed a mixed picture in terms of fold increase and exhibited generally low gross and net conversions.

Overall, no clear trends could be identified either in terms of the type or position of the substituent. An analysis of gross and net conversions as well as fold increase also showed no clear trends with respect to substrate $pK_a$ or Hammett substituent constants [23]. Thus, the results must arise from a complex interplay of stereoelectronic interactions in the active site of the enzyme. However, as the crystallographic structure of the active enzyme is currently unknown, it is not possible to infer specific interactions.

## 2.2. Triethylsilylation of Aliphatic Alcohols

The condensation of a range of alcohols was then investigated under identical reaction conditions to the phenols. Initially, 1-octanol was tested but gave no conversion above the control reaction even after 192 h (Figure 2, Table S1 in Supplementary Information). This result contrasts with the results from the same protein bearing a hexahistidine affinity tag that gave a net conversion of 86% after just 72 h [20], but is consistent with the observations from a more recent report showing that the exchange of this tag for a streptavidin affinity tag resulted in diminished activity [21]. Similar results are reported with *E*-3-penten-2-ol, with essentially no activity above baseline.

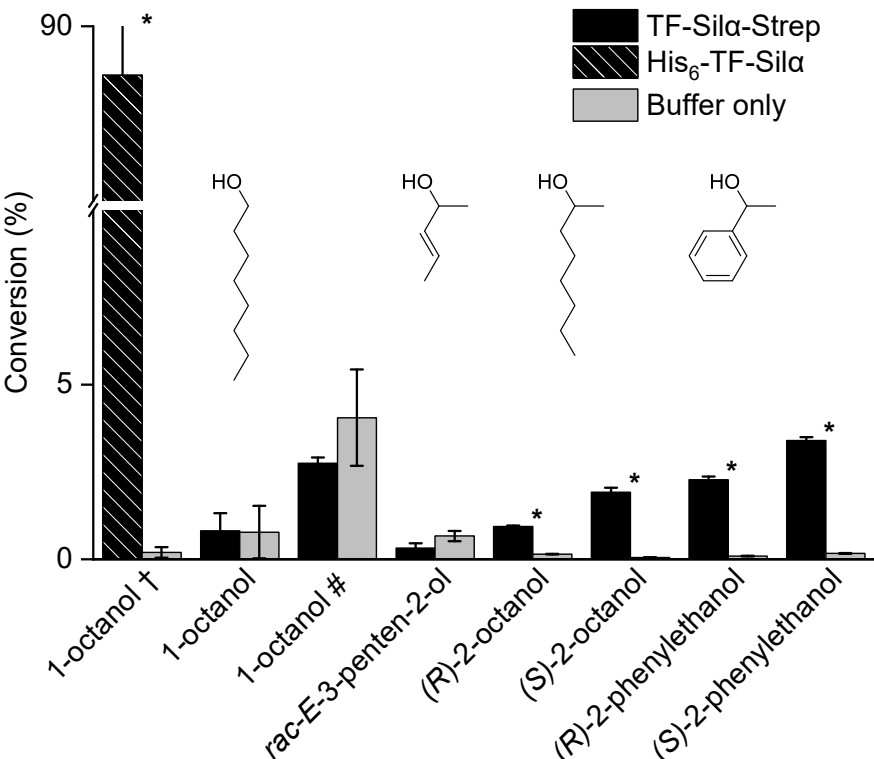

**Figure 2.** Graph of percentage conversions of alcohols to the corresponding silyl ethers after 72 h. The error bars indicate standard deviations. A one-tailed Student's *t*-test assuming unequal variance was performed, comparing each enzyme to its control. Comparisons resulting in a *p* < 0.05 were deemed to be significant and are marked with *. The † indicates data for the hexahistidine-tagged enzyme taken from [20]. The # indicates a reaction time of 192 h. The structures of the alcohols are also shown inset.

Two pairs of chiral alcohols, 2-octanol and 1-phenylethanol, were also investigated in this survey to assess any potential enantioselectivity displayed by the enzyme. Notably, despite the lack of significant conversions with 1-octanol or (racemic) 3-penten-2-ol, both enantiomers of 2-octanol did result in significant conversions above that of the control reactions. These conversions were extremely low at 0.8% for the *R*-enantiomers and 1.9% for the *S*-enantiomers (net), but were statistically significant. Furthermore, these values represented 6.3- and 38.8-fold increases in conversion compared to the controls. This result demonstrates that the enzyme preferentially catalysed the condensation of the *S*-enantiomer. For the phenylethanols, slightly higher gross conversions of 2.2% and 3.2% (net) were respectively observed for the *R*- and *S*-enantiomers. However, the 25.5- and 20-fold increases compared to the control indicated a lower level of enantioselectivity.

From these results, only very general trends regarding the reactivity could be drawn. The aliphatic alcohols were overall less reactive than the phenols in both enzymatic and control reactions, which may be related to their Brønsted acidity. No clear insights were gained when comparing the individual aliphatic alcohols. Indeed, the results superficially appear to be contradictory. Large hydrophobic alkyl chains such as those presented by the 2-octanols were accepted (albeit with low conversion), but 1-octanol was not. Likewise, both 1-phenylethanols were accepted, but *rac-E*-3-penten-2-ol was not. Substrates containing an aromatic ring appeared to have higher conversions (the phenols and phenylethanols), which suggests preferential binding with the enzyme binding site. This observation appears consistent with earlier computational modelling that indicated a possible cation–π interaction between the ring and a nearby arginine residue [20]. Conversely, the very low level of activity of Silα with the aliphatic substrates could simply be due to the lack of any attractive interactions with these otherwise largely unfunctionalized molecules.

### 2.3. Screening of Reaction Media

The current procedure for silyl ether condensation with Silα is based on previous work that utilises octane as the reaction medium [20]. However, this solvent greatly limits the range of substrates that can be applied, due to their low solubility. Thus, we tested a selection of solvents in an attempt to address this shortcoming. Several polar solvents were chosen (ethyl acetate, tetrahydrofuran, 1,4-dioxane and diisopropyl ether) and toluene was used as a model aromatic solvent. *m*-Methoxyphenol was chosen as the substrate, as the above results showed a moderate conversion, allowing for leeway to demonstrate both increases and decreases in conversion. As before, the reactions were carried out using lyophilised enzyme at 75 °C and analysed by GC-MS after 72 h. In line with earlier results, some silyl ether product was formed in all cases, even when the enzyme was omitted (Figure 3, Table S2 in Supplementary Information). Increased conversions above that of the control reactions were observed with the nonpolar solvents *n*-octane and toluene, with both giving similar net and gross conversions. In contrast, no net conversion was observed in any of the polar solvents.

To gain further insight, an analysis was performed to correlate net conversion with solvent polarity. For this purpose, the solvent polarity was quantified using empirical "normalised electronic transition energies" ($E_T^N$) as described by Reichardt [24,25]. This measure is based on the energy of the π–π* transition of a solvatochromic dye in a solvent and adjusted to a range between 0.0 (for tetramethylsilane) and 1.0 (water). Notably, this analysis showed that the silanol condensations were sharply demarcated, whereby solvents with $E_T^N > 0.1$ gave essentially no conversions over the control experiments (Figure 3, Table S2 in Supplementary Information).

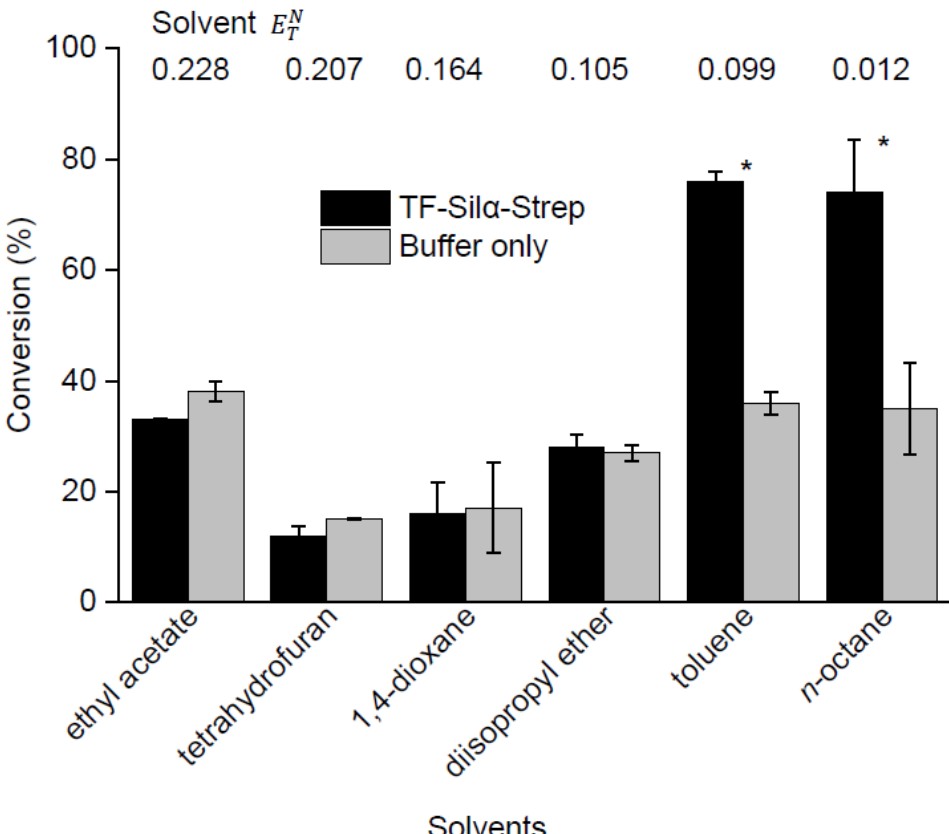

**Figure 3.** Graph showing percentage conversions of *m*-methoxyphenol and triethylsilanol condensations in a range of solvents after 72 h. A one-tailed Student's *t*-test assuming unequal variance was performed, comparing each enzyme to its control. Comparisons resulting in a $p < 0.05$ were deemed to be significant and are marked with *. The normalised electronic transition energies for the corresponding solvents are shown inset.

It has long been known that when using lyophilised enzymes in organic solvents, nonpolar solvents generally give a higher level of activity because the more polar (or hydrophilic) solvents remove the essential aqueous monolayer that surrounds the lyophilised enzyme and causes protein denaturation [26,27]. The results presented here conform to this postulation and therefore suggest that catalysis requires a correctly folded (not denatured) enzyme; that is, the results are not solely due to simple acid–base catalysis afforded by the presence of acidic or basic amino acid residues. Indeed, acid–base catalysis via a classical $S_N2$ mechanism would have been expected to produce higher conversions in polar aprotic solvents, which is not the case here.

As attempts to increase the polarity of the reaction media with a single solvent were unsatisfactory, mixtures containing increasing proportions of 1,4-dioxane in *n*-octane were assessed. 1,4-Dioxane was selected due to its lower polarity (and is therefore less likely to remove the aqueous monolayer) and boiling point (101 °C) that more closely matches that of *n*-octane. In general, the net conversion fell with increasing proportions of polar solvent (Figure 4, Table S3 in Supplementary Information). Reactions with 30% 1,4-dioxane gave no statistically significant difference between the enzyme and control conversions, presumably due to the removal of structural water crucial to enzyme function, as mentioned above.

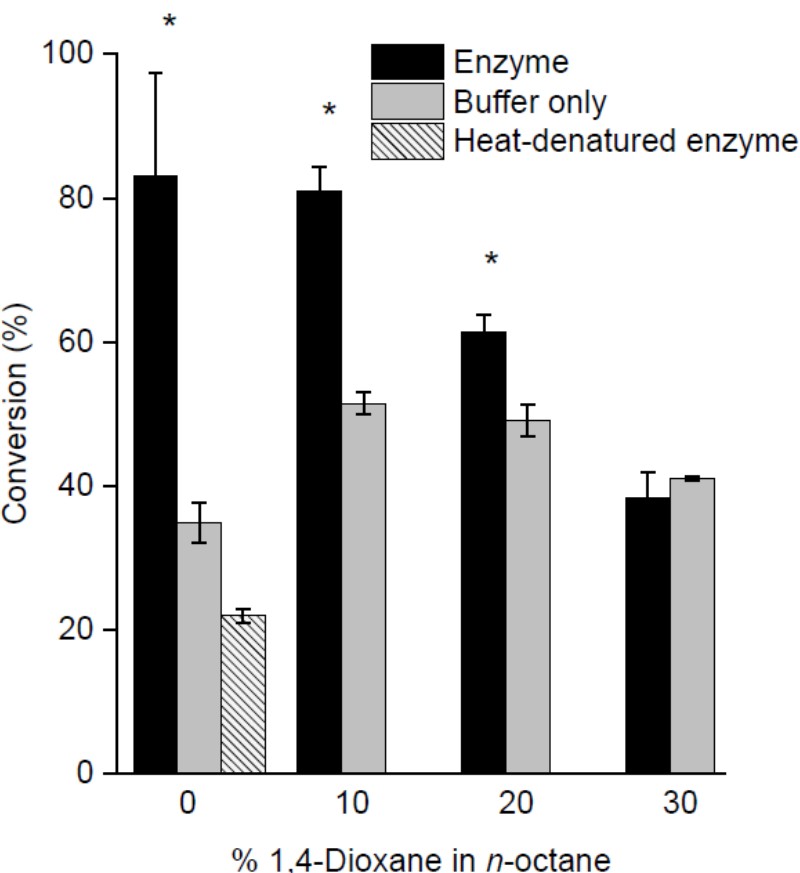

**Figure 4.** Graph showing percentage conversions of *m*-methoxyphenol and triethylsilanol in varying percentages of 1,4-dioxane in *n*-octane (solvent mixtures) after 72 h. A one-tailed Student's *t*-test assuming unequal variance was performed, comparing each enzyme to its control. Comparisons resulting in a $p < 0.05$ were deemed to be significant and are marked with *.

As a final control experiment, the enzymatic condensation of the *m*-methoxyphenol in neat *n*-octane was carried out using a sample of enzyme that was denatured by heating prior to lyophilisation, to confirm that the enzymatic condensation was indeed the result of specific catalysis and not simply the presence of the polypeptide chain. It was found that the heat-denatured enzyme gave a conversion that was no better than the negative control where no enzyme was added (Figure 4).

## 3. Materials and Methods

### 3.1. Materials and Equipment

All solvents and reagents were purchased from either Sigma-Aldrich (now Merck), VWR or Fisher Scientific. All solvents used were supplied as anhydrous, except *n*-octane and isopropyl ether, and used without further purification. Authentic samples of all product silyl ethers were prepared by conventional synthetic silylation of the corresponding alcohol or phenol with chlorotriethylsilane under basic conditions [8] (see Supplementary Information), and were used as standards for the GC-MS analysis. The enzyme was heterologously produced in *E. coli* as previously described [21].

The condensation reactions were carried out in crimp-sealable 8 mm vials (Chromacol C4008-741) that were heated and shaken on an Eppendorf Thermomixer 5350. GC-MS analyses were carried out using an Agilent 5975 Series MSD with the experimental parameters stated below (Table 1).

**Table 1.** Summary of GC-MS experimental parameters.

| Parameter | Setting |
| --- | --- |
| Instrument | Agilent 5975 Series MSD |
| Carrier Gas | 9.9995% ultra-high-purity helium |
| GC Inlet, Split | 240 °C, split flow 100 mL min$^{-1}$, split ratio 50 |
| MS Ionisation | Electron ionisation |
| GC Temperature Program | 50 °C (2 min)→240 °C (8.5 min) at 30 °C min$^{-1}$, 10.7 min total run time |
| GC Column | VF-5ht |
| Volume Injected | 1 μL |

### 3.2. Preparation of Lyophilised Enzyme and Matrix

The purified enzyme was buffer exchanged into the lyophilising buffer (100 mM KH$_2$PO$_4$, 100 mM K$_2$HPO$_4$, 20 mM KCl, pH 7) via PD-10 gravity-fed desalting columns in 2.5 mL batches (with multiple columns used in series for large batches). The protein concentration was adjusted to 5 mg mL$^{-1}$ and 18-crown-6 was added to a 0.04 mM concentration in the final solution. Aliquots of 100 μL of the enzyme solution were placed in the glass vials, flash frozen by plunging them into liquid nitrogen, and then lyophilised. For the negative control where the enzyme was omitted, 100 μL aliquots containing only lyophilising buffer with 18-crown-6 were flash frozen and lyophilised. For the control experiment using heat-denatured enzyme, the 100 μL aliquots in the glass vials were heated at 95 °C for 30 min and allowed to cool to ambient temperature before being flash frozen and lyophilised.

### 3.3. Enzymatic Condensation Reactions

A stock solution was first prepared by mixing the desired alcohol (1.26 mmol, 420 mM) and triethylsilanol (6.33 mmol) in the desired solvent (3 mL). 100 μL of this mixture was added into each vial containing the lyophilised enzyme (see above) and the vial crimp sealed. Reaction vessels were then heated at 75 °C while shaking at 650 rpm. At the desired time point, hexane (1 mL) was added, and the mixture centrifuged (17,000× *g*, 10 min) to separate the solid matter. 1 mL of the supernatant was transferred to a clean vial and subjected to GCMS analysis. Each reaction was performed in triplicate, and error bars presented in the figures refer to the standard deviation of these three independent data point measurements. For quantification of conversion rates, the GC-MS was first calibrated (Figures S2–S13 in Supplementary Information) using the synthetically prepared standards.

### 4. Conclusions

In summary, a survey into the reactivity of TF-Silα-Strep with a selection of aromatic and aliphatic alcohols has been conducted. In general, these enzymatic silyl condensations show a preference for aromatic alcohols (i.e., phenols) over aliphatic alcohols, as evidenced by much higher net conversions. However, greater improvements in conversions were achieved with the chiral aliphatic alcohols, as quantified by the fold increase, since the uncatalysed reactions with these alcohols showed proportionally much lower levels of product formation. In addition, Silα has a preference for the *S* enantiomers of the substrates tested, albeit with only low levels of conversion. A subsequent survey of solvents for this reaction showed that condensation could be effected only in nonpolar solvents ($E_T^N > 0.1$), though the addition of a small amount of polar solvent (up to 20% 1,4-dioxane) was tolerated.

As the main aim of this study was to investigate the substrate scope of Silα, these results are not currently synthetically useful except perhaps in a few cases (e.g., phenol, *p*-methoxyphenol). Nevertheless, this intrinsic activity offers a good starting point for directed evolution [28] into expanding the substrate scope of Silα. These results thus lay the foundation for future exploitation of Silα's chemo- and enantioselectivity toward a "silyl etherase" for practical biocatalysis. Additional work in this direction would benefit

from screening a wider variety of aromatic hydroxyl groups, competition experiments with two alcohols and further elucidation of Silα's enantioselective capabilities through screening racemic alcohol mixtures.

**Supplementary Materials:** The following are available online at https://www.mdpi.com/article/10.3390/catal11080879/s1, Figure S1: Graph of percentage conversions of phenols to the corresponding silyl ethers after 72 h. Table S1: Percentage conversion, net enzymatic conversion and conversion enhancement for the condensation of aromatic alcohols and triethylsilanol after 72 h. Table S2: Percentage conversion, net enzymatic conversion and conversion enhancement for the condensation of *m*-methoxyphenol and triethylsilanol in various solvents after 72 h. Table S3: Percentage conversion, net enzymatic conversion and conversion enhancement for the condensation of *m*-methoxyphenol and triethylsilanol in various mixtures of 1,4-dioxane and *n*-octane. Figures S2–S13: Calibration graphs of area under the peak corresponding to silyl ethers in the GC-MS trace against concentration, compound characterisation data for the authentic silyl ether products.

**Author Contributions:** Conceptualisation, S.Y.T.D. and L.S.W.; methodology, E.I.S., B.L., R.S., S.A.C. and S.Y.T.D.; validation, E.I.S. and C.S.E.; formal analysis, E.I.S., B.L., P.Q. and L.S.W.; investigation, E.I.S., C.S.E., B.L. and S.A.C.; resources, S.A.C. and S.Y.T.D.; data curation, E.I.S., C.S.E. and L.S.W.; writing—original draft preparation, E.I.S.; writing—review and editing, C.S.E., P.G.T., P.Q. and L.S.W.; visualisation, E.I.S. and C.S.E.; supervision, L.S.W.; project administration, L.S.W.; funding acquisition, P.G.T., P.Q. and L.S.W. All authors have read and agreed to the published version of the manuscript.

**Funding:** This research was funded by the UK Engineering and Physical Sciences Research Council (research grant EP/S013539/1 and graduate studentship EP/M506436/1 to E.I.S.), the UK Biotechnology and Biological Sciences Research Council (research grant BB/L013649/1 and graduate studentship BB/J014478/1 to S.A.C.) and the Tertiary Education Trust Fund of Nigeria (graduate scholarship to C.S.E.). The GC-MS analysis was performed in the analytical facilities of the Manchester Synthetic Biology Research Centre for Fine and Specialty Chemicals (SYNBIOCHEM), funded by grant BB/M017702/1.

**Data Availability Statement:** The datasets generated during during the current study are available from the corresponding author on reasonable request.

**Conflicts of Interest:** The authors declare no conflict of interest. The funders had no role in the design of the study; in the collection, analysis, or interpretation of data; in the writing of the manuscript or in the decision to publish the results.

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
