# Peer review of "Biocatalytic Silylation: The Condensation of Phenols and Alcohols with Triethylsilanol"

_catalysts, doi:10.3390/catal11080879_

Round 1

Reviewer 1 Report

Please see attached document for associated edits and comments.

Reviewer 2 Report

In their submission to Catalysts entitled "Biocatalytic Silylation: The Condensation of Phenols and Alcohols with Triethylsilanol", Sparkes et al. describe an interesting methodology of condensation of alcohols with triethylsilanol  catalyzed by the enzyme Silicatein-α. This enzyme catalyzed the triethylsilylation of various substituted phenols an aliphatic alcohols  for a reaction time of 72 h. Reaction conditions were also optimized and best results were obtained in when using non-polar solvents, particullarly a mixtrure of 1,4-dioxane and n-octane. The paper is well written and conclusions are supported by experimental evidences. This manuscript is of interest of the readership of Catalysts and deserved to be published as a an Article. Therefore, I would suggest accepting this paper after having taken into account the following points: 

1) A kinetic study using a phenol and an aliphatic alcohol needs to be included.

2) As the enzyme shows a preference for phenols, a competition experiment with a phenol and an aliphatic alcohol should be interesting.

Round 2

Reviewer 2 Report

The authors have addressed some of th comments proposed by the reviewers and the manuscript can be accepted at this stage.